# Effects of Game-Enhanced Supplemental Fraction Curriculum on Student Engagement, Fraction Knowledge, and STEM Interest

**Jessica H. Hunt** [1,*] , **Michelle Taub** [2] , **Matthew Marino** [2] , **Alejandra Duarte** [1] , **Brianna Bentley** [1] , **Kenneth Holman** [2] **and Adrian Kuhlman** [1]

1 College of Education, North Carolina State University, Raleigh, NC 27606, USA; aduarte2@ncsu.edu (A.D.); brbentle@ncsu.edu (B.B.); akuhlma@ncsu.edu (A.K.)
2 College of Community Innovation and Education, University of Central Florida, Orlando, FL 32816, USA; michelle.taub@ucf.edu (M.T.); matthew.marino@ucf.edu (M.M.); kenneth.holman@ucf.edu (K.H.)
* Correspondence: jhunt5@ncsu.edu

**Abstract:** People with disabilities are underrepresented in STEM as well as information, communication, and technology (ICT) careers. The underrepresentation of individuals with disabilities in STEM may reflect systemic issues of access. Curricular materials that allow students to demonstrate their current fraction knowledge through multiple means and provide opportunities to share and explain their thinking with others may address issues of access students face in elementary school. In this study, we employed a sequential mixed-methods design to investigate how game-enhanced fraction intervention impacts students' fraction knowledge, engagement, and STEM interests. Quantitative results revealed statistically significant effects of the program on students' fraction understanding and engagement but not their STEM interest. Qualitative analyses revealed three themes—(1) Accessible, Enjoyable Learning, (2) Can't Relate, and (3) Dreaming Bigger—that provided contextual backing for the quantitative results. Implications for future research and development are shared.

**Keywords:** mixed methods; serious games; supplemental curriculum; fractions; equity; disability; technology

Individuals with disabilities possess unique skills and talents that are important for Science, Technology, Engineering, and Mathematics (STEM) fields and Information, Communication, and Technology (ICT) careers [1]. For example, individuals with high incidence disabilities (HID), such as autism or learning disabilities (LD), have the ability to persist in conceptualizing outcomes and solutions to complex problems alongside procedural knowledge and skills [2]. Individuals with disabilities also possess strong intrapersonal skills and a sustained attention to detail that can also be important assets to aspiring STEM professionals [3]. These attributes uniquely position individuals with disabilities to make innovative and potentially transformative contributions to STEM and ICT fields.

People with disabilities are underrepresented in STEM as well as information, communication and technology (ICT) careers. For example, 38% of individuals aged 16–64 with disabilities were employed in 2022, compared with 77% of their peers without disabilities [4]. There are multiple reasons for this, including (1) limited encouragement to participate in STEM careers, (2) a lack of role models in STEM, and (3) cultural biases and stigma negatively impacting student perceptions of inclusion in STEM [5].

The underrepresentation of individuals with disabilities in STEM may reflect systemic issues of access. For example, the ability for students with HID to have access to and agency over which tools or representations they use to learn or express knowledge is highly beneficial [6]. These students also benefit from content and learning experiences that have multiple solution strategies that allow them to access and use their prior knowledge [7,8].

Unfortunately, much of the current research and programming around supplemental interventions do not consider designing for access and opportunities to engage in and to express unique reasoning.

Game-based mathematical interventions may be a powerful way to create access and opportunity in STEM careers by improving engagement and learning outcomes in content foundational to STEM, such as fractions [9–12]. Meta-analyses of research on gaming illustrates the potential of games to improve content accessibility through sandbox play and problem solving, which provide opportunities to build self-regulation and to explore content in ways previously inaccessible for students with disabilities [13]. Furthermore, video games that include the UDL framework (e.g., digital tools located in the user interface or information available using multiple modalities) have been empirically shown to eliminate gaps in knowledge between students with and without disabilities in middle-school science classes [6]. However, there is a pressing need to determine if the same results will occur in mathematics.

Therefore, the purpose of this paper is to explore how a game-enhanced supplemental fraction curriculum impacts student engagement and fraction knowledge in elementary school mathematics. A sequential (QUANT—QUAL) mixed-methods design is employed to investigate how game-enhanced fraction intervention impacts students' fraction knowledge, engagement, and STEM interests. The research questions are:

a.  Are there significant differences between students' initial and post-fraction schemes and STEM interest before and after participating in a game-enhanced intervention?
b.  Are there different levels of self-reported student engagement during the game-enhanced fraction intervention? Does engagement differ based on teacher/classroom assignment and/or disability status?
c.  What are students' perspectives about their engagement, STEM interest, and fraction knowledge after participating in game-enhanced fraction intervention?

## 1. Literature Review

### 1.1. Fraction Knowledge and Students with HID

Mathematics education research in fractions has documented for decades the challenging nature of fractions [14–22]. In a similar way, special education research has contributed to our understanding of the unique strengths and needs of elementary students with HID [23–26]. For example, we know that longitudinal growth patterns begin at earlier stages and these beginnings are more prevalent (and disproportionate) among students with HID [8,24]. In fact, students with HID were described as having inflexible thinking with regard to fraction concepts and estimation over time, and they exhibited significant differences with additive and multiplicative operations on fractions and in understanding fractions as two whole numbers as opposed to a multiplicative coordination [24–26]. These studies suggest that students with HID require opportunities to expand their whole number knowledge with fractional units.

At the same time, special education has also researched and advocated for instructional design considerations to promote access for students with HID to fractions. Specifically, studies suggest that using a carefully designed sequence of tasks and/or representations is beneficial to students with HID [27,28]. More recent work [23,25] uses fractions as measures, or lengths on a number line, to replicate and extend the structuring of whole numbers, with iterated unit fractions repeated to produce non-unit fractions. This process combines whole numbers with rational numbers and draws upon students' prior knowledge of whole numbers to conceptualize fractions. Yet, much of this research recommends teaching students to replicate teacher strategies, modeled representations, and verbalized explanations as a way to access and learn the content.

Although teacher-modeled instruction does show consistent, positive effects on specific aspects of fraction concepts, it rarely considers the students' own sense making [29], places notions of responsiveness on the individual student [30] and disallows students access to their own viable ways of thinking [30,31]. These environments also show limited

engagement and reduced attendance by students with HID, possibly due to inaccessibility, limited use of their own ideas [8], or boredom [3]. Therefore, more research and development of curricular materials is needed that (a) allows for students to demonstrate their current fraction knowledge through multiple means and (b) gives students opportunities to share and explain their thinking with others.

### 1.2. Game-Enhanced Interventions

Game-based learning environments are effective classroom tools because they can be used to (1) supplement core curriculums and (2) serve as an instructional tool for STEM topics. Additionally, games can serve as additional support for students with HID in the mathematics classroom. For example, game-based learning environments have been developed and shown to improve cognitive function and build and internalize knowledge for individuals with HID [32–35].

Further, many meta-analyses have been conducted that have investigated the impact of games across instructional domains, including mathematics [36–41]. In general, these studies have found conflicting results on the specific effects of games on learning outcomes when including moderators in the analysis, with games being more effective with respect to learning outcomes in some domains but not others. However, the more recent meta-analyses, which have focused specifically on game-based STEM instruction [39,40], show more promise. Specifically, [40] report results consistent with [39], who found that digital game-based instruction had a positive impact on learning in science, mathematics, and engineering. Digital games including immediate cognitive feedback improved students' mastery of problem-solving tasks within a game [42]. Summaries of gaming research have identified the potential of games to enhance STEM content accessibility and increase collaborative problem-solving and exploration. Nevertheless, specific reports of the impact of games on elementary students' mathematics knowledge and their engagement or how games might bolster interest in pursuing STEM and ICT careers is not yet well understood. Thus, more research is needed to investigate how novel game-enhanced curricula based specifically on fraction knowledge might positively impact outcomes for elementary-aged students.

## 2. Theoretical and Conceptual Frameworks

This study draws upon the theoretical framework of Universal Design for Learning and the conceptual frameworks of Learning Trajectories and Scheme Theory, as shown in Figure 1. Each framework is explained below.

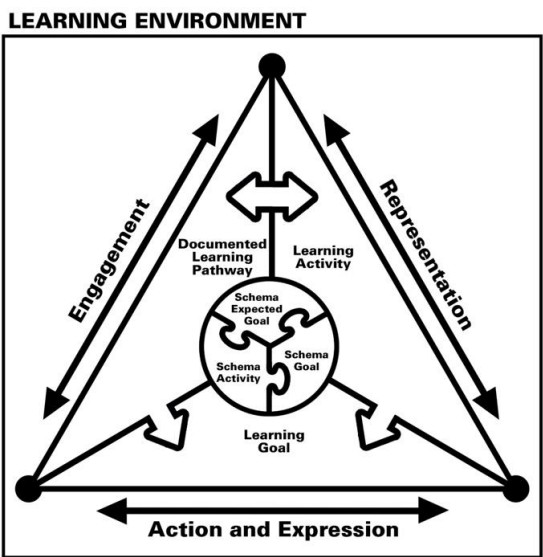

**Figure 1.** Theoretical framework.

### 2.1. Universal Design for Learning

The Universal Design for Learning (UDL) framework is an educational approach that aims to ensure that all learners can access, participate in, and progress in learning activities regardless of their abilities, backgrounds, or disabilities. The UDL framework provides educators with a set of guidelines to create inclusive learning environments and to design flexible learning experiences that meet the diverse needs of all learners. According to [43], the UDL framework is based on three principles: providing multiple means of representation, multiple means of expression, and multiple means of engagement.

The guidelines and subsequent checkpoints of the UDL framework are interpreted as flexible guidelines rather than rigid prescriptions. Educators are encouraged to use the guidelines as a starting point to design learning experiences that meet the diverse needs of their learners. The guidelines are not intended to be followed strictly; rather, they offer a flexible framework that can be customized to the specific needs of each learner. For example, one of the guidelines is to provide multiple means of representation, which means providing information in different formats, such as text, images, and videos. One checkpoint related to this guideline is to provide options for perception. Educators can interpret this checkpoint by providing multiple options for learners to access information, such as closed captions, audio descriptions, and text-to-speech tools.

As noted above, agency students with HID have over the ways in which they engage, represent, and express their knowledge is necessary to promote long-lasting learning [44]. The primary focus of UDL is to design interventions that support learner variability. UDL demands that learning environments be intentionally planned so that they are accessible yet personally challenging for all learners. When planning for learner variability, intervention designers should consider specific considerations such as individual and group strengths, weaknesses, abilities, understanding of background knowledge, and motivation for participating in the learning activity. They use UDL to target specific methods and materials that will engage learners and provide multiple ways for students to gain information, build concepts, and express understanding (see Figure 2). In these ways, UDL helps intervention designers consider student-level variability as well as content accessibility.

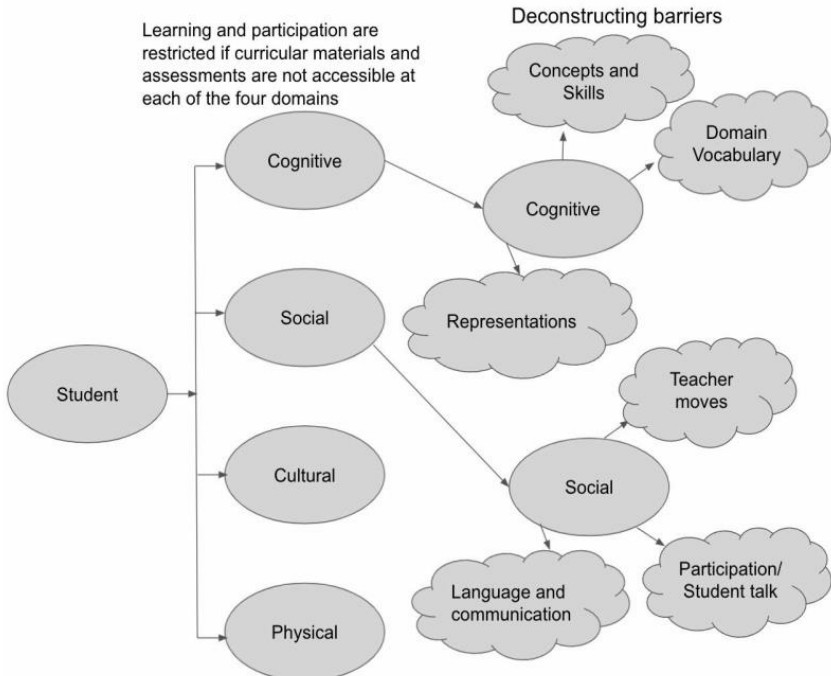

**Figure 2.** Universal design for learning considerations.

### 2.2. Learning Trajectories and Scheme Theory

The game-embedded curriculum was based on prior work [45,46] that established fraction-learning trajectories of students with mathematics disabilities and difficulties. Learning trajectories (LT) consist of a learning goal, documented pathways of possible development of a concept or idea, and activities designed to explicitly promote student reasoning [47]. Mental objects and actions underlie student thinking along each individual pathway. Students' mental activity and operations can be thought of as schemes. By schemes, we mean students' mental structures, containing three parts: a goal, an activity triggered by the goal, and an expected result of activity [48]. Schemes are rooted in radical constructivist theory, which states that each child actively constructs schemes through their own experience within an environment [48]. Individual experience gives students opportunities to engage in, notice, and reflect upon their unique problem-solving activity through a process that entails reconciling differences between expected and actual results of activity through reflection. This process affords students advances in their thinking, which constitute the possible pathways along the LT.

### 2.3. Theory-Backed Program Core Components

The four core components that comprise the intervention are based on the above theoretical and conceptual framing. Multiple means of engagement, representation, or expression support the design of the game's interface so that players can access each challenge in multiple ways. For example, an interactive learning environment motivates players by allowing them to customize the game based on their preferences. The player has a choice of flexible methods, materials, and analytical tools that they can use to employ individual strategies and ways of reasoning. Sandbox play supports players to create fractional quantities by partitioning, repeating, distributing, and coordinating units without high-stake repercussions. Carefully sequenced tasks are designed to bring forward concepts of fractions [49,50], the first four of which are defined in Table 1. These concepts are supported by the student actions of partitioning, iterating, and the coordination of partitioning and iterating into a group structure called splitting, the cognitive structure that underlies the concept of fractions as quantities. Because the design of the game and its predecessor [45,46] supports concepts [49], the design of the tasks, representations, and activities in the game are distinct from designs that employ task analyses or that promote the amassing of skills.

**Table 1.** First 4 game concepts.

| Concept | Definition |
| --- | --- |
| Unit fractions (1/n) | A measure of the whole that fits *n* times within the whole such that the the whole is *n* times as much of 1/n. |
| Partitive Fractions (m/n) | Iterating a given unit fraction (1/n) a few (*m*) times, not exceeding the *n/n* whole (i.e., m $\leq$ n), yields a *composite fraction* that is *m times as much as 1/n.* |
| Iterative Fractions (m/n) | Iterating a given unit fraction (1/n) a few (*m*) times exceeding the *n/n* whole (i.e., m $\leq$ n), yields a *composite fraction* that is *m times as much as 1/n.* |
| Reversible Fractions (m/n $\rightarrow$ 1/n $\rightarrow$ n/n) | Reversing the iteration supposedly used to create a composite fraction m/n by partitioning it into *m* parts to create 1/n and then "undo" the initial partitioning of the whole, which created 1/n, by iterating *n* times to make the *n/n whole.* |

Adaptation of thinking is supported by prompts that promote students' noticing and reflecting upon their actions on an object. Promoting noticing and reflection in students' thinking while solving problems is underutilized in instruction for students with HID [51–53]. Research suggests that students with HID rely upon suboptimal strategies

(i.e., guess and check) as they solve problems [54]. We contend that the use of such strategies is due to the limited opportunities students have to learn how to self-regulate within their own problem-solving actions. Students rely on ineffective regulatory processes and communication of reasoning because reflection is left unsupported in the midst of learning—students often require development of reflection with self-regulation [55]. In this way, prompts can be designed to support students' planning, goal setting, and reflection.

Finally, socially mediated learning occurs when teachers encourage students to explain and justify their mathematical thinking to others after engaging in a game-based task. The curriculum on which the reported game-enhanced program is based [45,46] used a think–pair–share structure so that students could think about a solution, rehearse thinking in pairs, and discuss their reasoning in a larger group. We extended the curriculum to integrate students' gameplay with opportunities for students to engage in mathematical reflection, explanation, and justification after they play the game through one of three pedagogical routines: a worked example, a game replay, or a number string. Worked examples relate to a concept that students are working toward within gameplay or a related skill that connects to a concept built up in gameplay. They help students to focus on the concept or meaning of a procedure by presenting problems solved correctly, in part, or incorrectly [56]. Game replays directly connect to a problem students encountered during gameplay in a particular world. They give students opportunities to re-create their gameplay, notice their strategy, and reflect upon the resulting quantity they created (see [57]. Number strings engage students in mentally solving a "string" of related equations. The repetitiveness encourages students to notice a particular mathematical concept or to make use of a particular strategy. When paired with conversation about how students solved a problem, specifically discussions that highlight efficient strategies and how the strategies work, number strings can support the abstraction of mathematical concepts [58].

## 3. Methods

### 3.1. Study Design

The current study examines a 5-unit, 30-lesson fraction curriculum with a video game embedded within it. The program is designed to maximize accessibility and engagement by providing fraction conceptual-understanding challenges rooted in authentic STEM and ICT careers. Careers include a wind turbine technician, a solar-panel installer, a fire inspector, a photogrammetrist, and a computer programmer. Each curriculum lesson has three parts: before-game previews, video game, and after-gameplay discussion activities (see intervention procedures). All components link to mathematics curriculum standards (e.g., National Council of Teachers of Mathematics, or NCTM).

A one-phase embedded sequential mixed-methods design [59] was used to address the research questions. In this type of design, one data set, in this case, the qualitative, provides a supportive secondary role in the study, based primarily on the quantitative data. Students' fraction schemes/concepts [60] and STEM interest [61] were measured before and after the 30-lesson program. Students' self-reported engagement [62] was measured after each game world and associated lessons. Next, semi-structured focus groups were conducted with students at the conclusion of the curriculum. Finally, results from both sets of data were compared and merged for a final interpretation.

### 3.2. Participants and Setting

The study was conducted with six 4th and 5th grade teachers and their students ($n$ = 133) at two different schools in the southeastern United States. Both schools included students with intersecting identities in terms of race, language, and disability. The intervention occurred in Tier 1 settings, which commonly includes 15–25 students with one teacher. The interventions in each school took place over nine weeks, which is considered best practice in terms of time period for technology-based interventions [63]. Demographic information for the six participating teachers and their students is given in Table 2.

**Table 2.** Demographics.

| Students | | | | |
|---|---|---|---|---|
| **School** | **Grade** | **Gender** | **Race** | **Disability Status** |
| 1 (44%) | 4th (43%) | Female (41%) | Hispanic (28%) | Yes (16%) |
| 2 (56%) | 5th (57%) | Male (59%) | White (32%) | No (84%) |
| | | | African American (21%) | |
| | | | 2 or more races (19%) | |

Teacher Development

Teachers attended four half-day training sessions on the implementation of the supplemental curriculum. Day one opened with the purpose of the study, the logic model, and the target population. Over the second and third days, teachers used the game and sample student gameplay to deepen their understanding of how the core program components are used to bolster student learning. On the final day, a curriculum guide was given to teachers to drive delivery of the intervention, and teachers practiced using the resource through role-playing in small groups, rotating between teaching roles and student roles. Teachers also engaged in the after-game tasks, discourse, and talk moves to facilitate a sample student conversation. Finally, training on how to administer the study's measures was given.

*3.3. Data Sources and Measures*

3.3.1. Fraction Knowledge

Fraction knowledge was measured before and after the intervention using the Test of Fraction Schemes [60]. This test is a 12-item measure of the effects of the intervention on students' fraction conceptions. Original numerical values and representations were retained in both the pre- and the post-test. Tests were paper-and-pencil-based. Internal consistency reliability for the test was reported as 0.70; criterion-related validity was reported as 0.58 ($p < 0.01$). The test was group-administered and included six items that measured the mental operations students use to construct fractions (i.e., two items on partitioning, two items on iterating, and two on splitting) and six items on overall fraction concepts (e.g., unit fraction, partitive fraction, iterative fraction).

3.3.2. STEM Interest and Student Engagement

We used the Upper Elementary School (4–5) and Middle High School (6–12) Student Attitudes Toward STEM (S-STEM) [61] and the Engagement in Science Learning Activities [62] surveys to measure changes in students' self-reported STEM interests before and after the intervention and self-reported engagement after each game world (five total measurements). To supplement the quantitative data, we also measured STEM interest and student engagement with semi-structured focus-group interviews with a subsample of 24–36 students at the conclusion of the study. Each measure is described below.

**STEM interest**

The S-STEM was developed as part of a US National Science Foundation (NSF)-funded research program and measures students' confidence and self-efficacy in STEM subjects, 21st-century learning skills, and interests in STEM careers. It contains 56 items across six constructs: math attitudes (8 items), science attitudes (9 items), engineering and technology attitudes (9 items), 21st-century learning attitudes (11 items), interest in STEM career areas (12 items), and 7 "About You" items that measure short-term expectations for course success and exposure to STEM careers. We used the math attitudes and the interest in STEM career areas for our study. Responses are supported by a 5-point Likert scale, with response options ranging from "strongly disagree" (1) to "strongly agree" (5). Higher

scores reflect the greater perceived value by participants. Cronbach's α of the S-STEM ranged from 0.84 to 0.86 for the grade 4–5 subscales.

**Self-reported student engagement**

The Engagement in Science Activities survey was written for use with 10- to 14-year-old students immediately after an in-class or informal activity. It is used to measure a student's cognitive, behavioral, and affective engagement. Participants respond on a Likert-type scale ranging from 1 (YES!) to 4 (NO!). "Because no particular assumptions are made about a task structure other than there is a particular task that should have been completed (p. 1)", valid inferences can be made regarding overall mathematics engagement during an activity using responses across all of the items. Both Cronbach's α and the polychoric coefficients yielded acceptable reliability when using all eight scale items (0.80 and 0.85, respectively). Exploratory factor analysis from an SEM bifactor model yielded a satisfactory fit (CFI = 0.992; TLI = 0.982; RMSEA = 0.069).

### 3.3.3. Student Focus Groups

Student focus groups were held at the conclusion of the study to gauge students' perspectives on their fraction knowledge, interest in STEM, and overall experiences in the game-enhanced curriculum. Six separate focus groups were held with a purposive sampling of four to six students from each teacher's classroom. Purposive sampling was used to ensure that the makeup of each group of students resembled the makeup of the entire class in terms of disability status, race, and gender. Teachers aided in student selection. The focus groups were held in person using semi-structured questions and lasted anywhere from 20 to 30 min. Participants were not compensated for their time. Focus-group questions included asking students about what they liked and did not like about the game, what they liked and did not like about the after-game tasks, what they did and did not like about the preview activities, and the extent to which they felt like they 'became' the Bunny as they played the game and participated in the curriculum. See the appendix for the full list of questions and protocol.

### 3.4. Intervention Procedures

Teachers used the program's curriculum guide to administer the supplemental lessons for 35 min a day, 3 days a week, for 9 consecutive weeks. For each lesson, teachers enacted a 5-min preview, allowed 10–15 min of student gameplay, and facilitated a 15–20 min after-game task (i.e., a number string, a game replay, or a worked example). Previews were video-based and were presented with one or two discussion questions. For example, one preview showed students how wind turbines are constructed and generate power, while another video showed students the location and prevalence of windmills across the country and in the students' state and counties. The previews were designed to invite student elaboration, conversation, and questions about a STEM or ICT career depicted in the game.

After the preview, students were invited to engage with the embedded universally designed video game for 10–15 min. The game is a five-world, sandbox-play puzzle game that presents fraction challenges along the learning trajectory. Students take on the role of "Bunny", a customizable character whose identity students can change according to their preferences. Fraction challenges range in concepts from unit fractions to partitive (i.e., non-unit) fractions to reversible fractions (e.g., using a non-unit fraction to produce a whole) to multiplicative concepts (e.g., taking a part of a part, distributing m fractional units over n whole units). Some challenges are more constrained to invite particular actions on objects (e.g., find the length of exactly one share of a rectangular length), while other challenges invite a range of approaches and ways of quantification. In every challenge, players are given a choice of tools that they can use to engage with the fraction challenges. The current version of the game is leveled; that is, students must successfully complete each challenge in order to advance to the next challenge, subworld, and game world. Gameplay is individualized and saves the student's progress along the way. As students play, different features of the character customizer "unlock".

After the game, students engage in a 15–20 min discussion with their classmates and their teacher through number strings and either a game replay (school 2) or a worked example (school 1). Game replay procedures are presented as think–pair–shares; students are given a challenge from the game world and are asked to recreate the strategy they used to solve it for four minutes. Next, students pair up and discuss their strategies with a partner for four to five minutes. They are given a sharing structure called talk and share. Sentence stems are provided to support students to both share (e.g., "my strategy was . . . "; "My strategy makes sense to me because . . . ") and listen to and question what is being shared (e.g., "I noticed that you said . . . "; "I had a question about . . . "). Finally, students come together as a whole group to share their strategies around the discussion focus for the lesson (e.g., partitioning or iterating strategy to make 1/n) and the overall learning goal for the world (e.g., unit fractions (1/n) are named for the number of times they repeat (n) to create some whole (n/n)). Procedures for worked examples are similar, with the exemption that students spend time in think-and-pair writing answers to questions posed about a correct, incorrect, or partially correct worked example. Procedures for number strings involved students being shown a problem that they were asked to solve in their heads and to indicate with a nonverbal symbol when they had a solution. The teacher then called on students to share their reasoning; teachers used a core representation (e.g., a number line) to illustrate the students' strategies. The process was then repeated with two to four additional problems; with each, teachers asked students if they could use their thinking from the previous problems to engage with the newly presented problem.

*3.5. Data Collection Procedures*

3.5.1. Quantitative Data Collection

Immediately prior to and immediately following the implementation of the program, teachers were asked to administer the Test of Fraction Schemes and the S-STEM survey to their students in their whole classroom settings in the morning hours immediately after the start of the school day. Collection took place in the first 35 min of school; students were given 30 min each to complete the pre- and the post-test. Teachers gave one test to each student, told students to write their name at the top of the tests, and to do their best work. Students were given no other direction. When requested, teachers read out loud test questions for students. Similar procedures were used for the math attitudes and interest in STEM (S-STEM) surveys, with a slight change in the direction given to students (i.e., to answer the questions using their own perspectives and experiences).

The Engagement in Science Activities survey was administered at the conclusion of each game world and its corresponding lessons. Survey administration occurred after each of the 5 game worlds within the first 35 min of the school day. Students were given five minutes to complete each survey and were asked to answer the questions using their own perspectives and experiences. Teachers then collected students' completed surveys and gave them to the researchers.

3.5.2. Qualitative Data Collection

One week after the conclusion of the final curriculum lessons, students were gathered into focus groups to inquire about their experiences of taking part in the game-enhanced curriculum. Students were dismissed from their teacher's classroom and led to a room in the school library. The room contained one rectangular table and several chairs. Two audio recorders were set up at either end of the table; a third recorder (i.e., quick-time audio via laptop) was positioned at the center of the table. After welcoming the students, the lead facilitators read a standard statement about the purpose of the focus group and that explained that students were free to participate as little or as much as they felt comfortable. Then, the facilitator asked each focus group questions (see Appendix A) one at a time. Secondary facilitators took detailed notes of the discussion, prompted the lead facilitator on time, sketched a drawing on the room with a description of each participant, and made a notation each time a participant made a comment to note trends. At times, the

facilitators would restate students' responses as a means of member checking or pose follow-up questions to gain elaboration on a response.

*3.6. Data Analysis Procedure*

We conducted three sets of data analysis—quantitative, qualitative, and the merging of both for a final interpretation.

### 3.6.1. Quantitative Data Analysis

To investigate changes in students' fraction schemes and STEM interest from pretest to post-test, we conducted paired-sample *t*-tests with time as the within-subjects factor to document the main effect of time (pre vs. post) on fraction scheme scores and survey scores for all students together (A repeated measures ANOVA was run for these data; however, the results violated Levene's test, so *t*-tests were employed to evaluate the first research question). Time of the test was the independent factor; fraction scheme score and STEM interest survey scores were the dependent factors. We also ran paired-sample *t*-tests for each school to understand how effects of the curriculum looked for students who were in classrooms at different schools (We acknowledge the nested structure of our data; however, limitations in sample size did not allow us to run these models).

To understand the differences in engagement between students in different teachers' classes and student disability status, we reviewed students' self-reported levels of engagement following their interaction with each of five curriculum units (i.e., each of the five game-world levels). Given the significant multicollinearity between participants' self-reported engagement scores across each game world, a *k*-means clustering analysis was performed to identify potential groups of students based on their self-reported engagement. Hierarchical clustering was first applied to participants' standardized engagement scores in all five worlds using Ward's method. Changes in the agglomeration coefficients (AC) and assessment of the resulting visualizations (i.e., elbow technique) were used to determine the ideal number of clusters [64]. After clusters were identified, Phi-Coefficient correlations were conducted to determine if students' cluster membership was significantly associated with whether the student reported having a disability and also by school assignment.

To understand student perspectives of their engagement, learning and STEM interest as a result of the supplemental intervention, we initially analyzed the focus-group data using concurrent rounds of open coding for each teacher for a within-case analysis [65]. After the data were transcribed and field notes and drawings were added, two research assistants cleaned the data and placed it in a spreadsheet. Data was chunked into smaller, more meaningful parts (i.e., talk turns). Descriptive coding [66] was used by each researcher to capture the experiences of students and their perspectives of the game-enhanced curriculum. Researchers independently labeled each talk turn with a descriptive title (i.e., code). Initial codes included initial content, after-task structure, group/partner share, multiple methods, missing instructions, problem wording, conversation length, prompts/hints, and learning fractions. Researchers met to compare and refine the codes and their meanings using collaborative work [67]; initial inter-coder agreement ranged from 85.19 to 100% for each teacher's data. This process was also helpful to build the trustworthiness of the data [68]. The descriptive coding process was then repeated to discern an additional two levels of codes to gain additional detail and specificity. Codes were then grouped (i.e., axial coding), and a category was identified for each grouping. Inter-coder reliability across codes was 93%. Next, a cross-case analysis was conducted to identify the shared experiences of students in the game-based curriculum across teachers [65]. Memos and collaborative work performed during the within-case analysis helped guide the cross-case analysis. That is, similar and contrasting evidence across cases was identified, compared, and contrasted to create a holistic description of the students' perspectives of the game-based curriculum. The combination of student focus-group data with the researchers' memos written throughout the analysis process helped to ensure the trustworthiness of the results.

3.6.2. Data Merging and Interpretation

After results from the quantitative and qualitative data were discerned, we merged the data analyses together. The quantitative data reveals differences in fraction knowledge and STEM interest before and after participating in the game-enhanced curriculum as well as process data on student engagement, while the qualitative data illustrates the perspectives of students' engagement, learning, and STEM interest after the program was completed. To gain a final interpretation of the data, trends in the data were identified, merged, and compared. First, researchers prepared a classical content analysis to quantify and identify trends in the qualitative data by school and for the group. Researchers counted how many occurrences comprised each grouping in the data set and divided the totals by the total number of all codes to obtain a percentage of time each subcategory and category were present in data for teachers in school 1 and school 2. Next, quantified data from the focus groups were merged with the change in fraction scheme scores, change in STEM interest pre- and post-surveys for each school, and engagement clusters for each school. Finally, commonalities or divergences across quantitative and qualitative data were compared, and *t*-tests were used to determine any significant differences in the number of codes appearing in each category and subcategory for each school to create a complete understanding of how the game-enhanced fraction intervention influences students' engagement, fraction knowledge, and STEM interest.

**4. Results**

*4.1. Quantitative Results*

4.1.1. Conceptual Understanding and STEM Interest

Our first research question asked if there are significant differences between students' initial and post-fraction schemes and STEM interest before and after participating in a game-enhanced intervention. A histogram confirmed the normality of the fraction test-score data. There was a statistically significant difference in fraction scores from pretest ($m = 2.74$, $sd = 2.34$) to post-test ($m = 3.51$, $sd = 2.53$), $t = -4.239$, $p < 0.001$. Cohen's d was 0.43, suggesting small- to medium-sized effects of the intervention for the group. The results suggest that students' conceptual understanding of fractions, as measured by the test (i.e., fraction schemes), increased at levels of practical significance.

Table 3 shows pre- and post-test means and standard deviations for students with different teachers in each school for the fraction knowledge test.

**Table 3.** Descriptive Statistics at the School Level for Fraction Schemes and STEM Interest.

|  | n | Pretest Mean | Standard Deviation | Post-Test Mean | Standard Deviation |
|---|---|---|---|---|---|
| | | | Fraction Schemes | | |
| School 1 | 56 | 3.39 | 2.66 | 3.93 | 2.95 |
| School 2 | 77 | 2.104 | 1.93 | 3.02 | 2.17 |
| | | | STEM Interest (Math Attitudes) | | |
| School 1 | 49 | 3.58 | 0.74 | 3.46 | 0.98 |
| School 2 | 76 | 3.50 | 0.96 | 3.36 | 1.16 |
| | | | STEM Interest (Interest in STEM) | | |
| School 1 | 49 | 2.52 | 0.65 | 2.12 | 0.79 |
| School 2 | 76 | 2.48 | 0.67 | 2.44 | 0.81 |

Statistically significant differences were found between pretest and post-test scores for students with teachers in school 2 ($t = -4.245$, $p < 0.05$) and for students with teachers in school 1 ($t = -2.483$, $p < 0.05$). Students with teachers in school 2 had slightly higher effect sizes (0.49) compared with results for the whole group (0.43), reflecting significant medium post-test gains for these students. Students with teachers in school 1 had lower

effect sizes (0.32) compared with results for the whole group (0.43), reflecting small but significant post-test gains for these students.

### 4.1.2. Math Attitudes and Interest in STEM

A histogram confirmed the normality of the survey data. There was no statistically significant difference in the average score from pretest ($m = 3.54$, $sd = 0.87$) to post-test ($m = 3.40$, $sd = 1.09$) ($t = 0.543$, $p > 0.0$) for the math attitudes survey for the group. There was also no statistically significant difference in average score from pretest ($m = 2.50$, $sd = 0.66$) to post-test ($m = 2.32$, $sd = 0.81$) ($t = -1.599$, $p > 0.05$) for the interest in STEM survey for the group.

When examining data at the school level, the results are mixed. Table 2 (above) also shows pre- and post-test means and standard deviations for students with different teachers in each school for the math identity and interest in STEM surveys. Both the means for math attitudes and interest in STEM decreased from pretest to post-test for students with teachers in school 1. However, statistically significant differences in average scores were found for Interest in STEM ($t = 2.659$, $p < 0.01$) but not Math Attitudes ($t = 0.927$, $p > 0.05$). Effect size for the significant change in Interest in STEM was 0.32, suggesting small to medium post-test losses in STEM interest for these students.

In school 2, both math attitudes and interest in STEM stayed relatively the same from pretest to post-test. That is, there were no statistically significant differences in the averages score for interest in STEM ($t = 2.659$, $p < 0.01$) or math attitudes ($t = 0.927$, $p > 0.05$) for students with teachers in school 2. This means that students' attitudes toward math and interest in STEM did not change significantly before and after participating in the game-enhanced curriculum.

### 4.1.3. Engagement

Our second research question asked if there are different levels of self-reported student engagement during the game-enhanced fraction intervention and if engagement differs based on teacher/classroom assignment and/or disability status. For our hierarchical cluster analysis, changes in the agglomeration coefficients (AC) and accompanying visualization (i.e., elbow method) suggested a 2- to 3-cluster solution, with the AC changing from 134.064 for a 4-cluster solution to 203.349 and 157.653 for the 2- and 3-cluster solutions, respectively. Therefore, *k*-means clustering was used to compare cluster centers and membership for the two- and three-cluster solutions. Given that the 2-cluster solution found large Euclidean distances between the final cluster centers, there were appropriate sample sizes for each cluster ($n > 15$), and all 5 variable center means were significantly different between the 2 clusters ($Fs < 0.001$), the 2-cluster solution was selected (see Figure 3).

Based on the two-cluster solution, we categorized the clusters as students who reported high levels and low levels of self-reported engagement. Cluster 1 ($n = 36$) included students who reported higher average standardized engagement scores following game play in each of the five in-game worlds within the program. Cluster 2 ($n = 24$) included students who reported lower average standardized self-reported engagement scores following game play in each of the five in-game levels.

Phi-Coefficient correlations were conducted to determine if there was a significant association between students' cluster membership and whether the student reported having a disability as well as which school the student attended. Given that cluster membership, disability status, and school-attended variables are all binary, categorical variables, Phi-Coefficient correlations were used to examine the relationship between (1) cluster membership (i.e., students' reported levels of engagement: 1 = high engagement; 2 = low engagement) and their disability status (1 = yes; 2 = no), and (2) cluster membership (same as above) and the school the student attended (1 = school 1; 2 = school 2). The results (see Figure 4) showed that there was no significant relationship between whether a student had a disability and their reported levels of engagement following game play; $\phi = -0.053$, $p = 0.683$. However, there was a significant relationship between the school the student

attended and cluster membership (i.e., whether they self-reported high or low levels of engagement); $\phi = -0.438$, $p < 0.001$. Students in School 2 were more likely to appear in the high-engagement cluster than students in School 1.

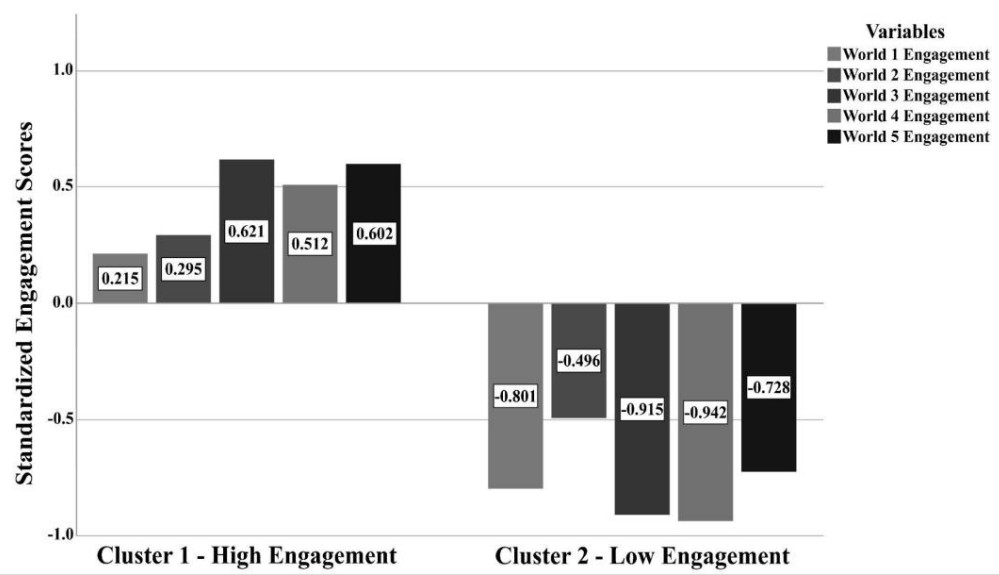

**Figure 3.** Final cluster centers for 2-cluster solution.

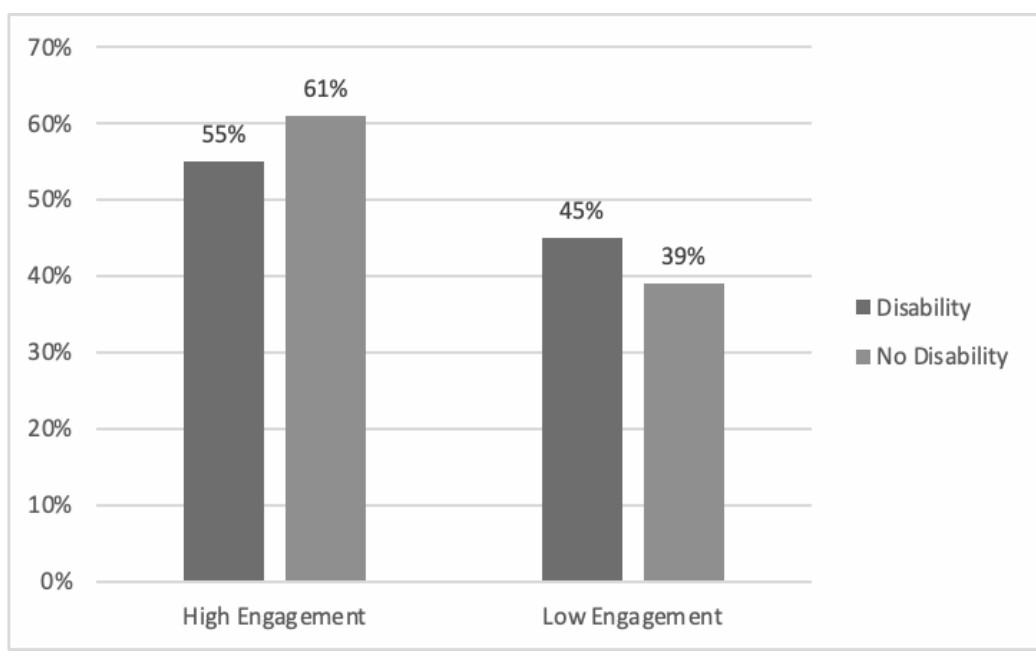

**Figure 4.** Distribution of students based on cluster membership and disability status.

### 4.2. Qualitative Results

Our qualitative analysis investigated students' perspectives about their engagement, STEM interest, and fraction knowledge after participating in game-enhanced fraction intervention. Three main categories emerged from the within- and across-case qualitative analysis: (1) Accessible, Enjoyable Learning, (2) Can't Relate, and (3) Dreaming Bigger. We unpack each category below.

4.2.1. Accessible, Enjoyable Learning

Students reported aspects of the launch activities, game, and tasks that they found helpful in the program. True to the nomenclature of the larger category, students' responses fell into three subcategories—(a) accessible, (b) enjoyable, and (c) promotes my learning—that are described below.

Accessible. Students described several aspects of the launch, game, and after-game tasks that were helpful to their learning. First, students described the game tutorial as being a helpful part of the launch activities and the game. Game tutorials were designed to showcase the tool functionality for each game world. One student commented that the tutorials were helpful "*because [they] would kind of give you a head start on what you were going to do*". Other students spoke of the tools themselves as supporting them to navigate the challenges smoothly. That is, students commented not only about the availability of certain tools in the game challenges but also about the ability to change the ways students engaged with them. One student explained, "*I like when you were on the settings and you could change how if you press a key it takes you to the cut tool or the drag tool. I was able to change the way I used the tools to numbers and also how I could access the bunny phone*". Finally, students described the multiple methods and solutions that were built into both the game challenges and the after-game tasks as accessible to their ways of reasoning. They spoke positively about the after-game tasks; in particular, with respect to the opportunity to showcase multiple methods or to come to more than one correct answer. A student explained, "*Like [T7, School 2] asked, 'Do you have something different?' for like three different types on each problem—so there were a lot of ways you could answer the problem*".

Enjoyable. Along with listing aspects of the program that students found accessible, they also spoke about enjoyable parts of the program. Features of the program that students found enjoyable generally involved STEM career videos and activities during the lesson preview, the context and characters of the game worlds, the ability to customize the Bunny, and the music involved. For example, one student described a preview video in World 2, saying, "*The video for example about the solar panel and how they get constructed, those were cool to watch*". Other students described enjoying the ability to customize the Bunny to their preferences. Two students commented:

> [Student 1] *It was like, I liked how you had to, like, change the character around. You could change their clothes and put different hair and outfits and glasses and all that. I liked that.*

> [Student 2] *I liked customizing the Bunny.*

Finally, students spoke positively about the background music of the game, stating that they often left the game running during times when they were completing other work.

Promotes my learning. Aspects of the program that promoted access and enjoyment appeared to coincide with opportunities for learning. For example, students described the STEM videos shown during the lesson previews, specifically, as part of the program that contributed to a positive experience. One student commented that these videos, specifically, helped them learn about the STEM careers depicted in the game. Other students agreed and added that the gameplay added to their understanding of STEM careers: "*I like how it teaches you how to do the job—and if you are older, you know what it's going to be like and if you like the job you can have it. You can try to get into that job when you're older*". Other students attributed gameplay to learning fractions. For example, one student commented, "*The game teaches you. I also like how w/fractions [the game] helps me understand it more, how to partition [to make] the parts, and make other fractions with them*". Other students connected fraction learning in the game to the after-game tasks, stating "*l kind of liked [the after-game tasks] because the questions . . . helped me with the game*". Finally, students spoke positively about the opportunity to share with a partner and/or the larger group when talking about the after-game tasks: "*Because sharing it with a group makes me understand that other person's thinking and the other person understanding what I'm doing*".

4.2.2. Can't Relate

Despite the aspects of enjoyability, accessibility, and learning, some students reported hindrances that were not beneficial in their experiences with the program. Specific areas discussed by students fell into three subcategories: (a) Not the math I am used to, (b) Not what I need when I need it, and (c) Not Bunny.

Not the math I am used to. At times, students described parts of the program as not consistent with what they knew mathematics to be or how they were used to experiencing mathematics. For example, in describing the tutorial videos, one student commented, "*Some of the videos, it didn't even show how to solve the problems—like in Gloria she was showing how solar panels work, but it didn't really show you how to do the math or solve the problems*". Another example of this sub-category rests in students speaking about the game in reference to the problem content or problem type/level of difficulty. One student was talking about the challenge of making the length of 1/n of a bar in the first world and explained, "*I didn't like it in the first world where you had to estimate the length of one of the parts—you had to guess . . . I did not like that*". Other students shared that they did not like or want to share with a partner or the larger group when engaging in the after-game tasks, explaining that they felt weird talking to other kids and did not want people to know what they thought or put as an "answer". Finally, other students talked about what they are used to as learners and/or how things are conducted in their regular mathematics classroom, explaining that they "*can't usually talk to neighbors when we do math; [teacher 1, school 1] doesn't let us work with people; sometimes they let us, most of the time they don't*".

Not what I need when I need it. Along with the program not aligning with some students' ideas about mathematics learning and teaching, other students pointed out times where the program did not meet them where they were. These examples generally involved the tools in the game, the problem content, or the amount of time students were given in certain parts of the program. First, students spoke about the gameplay in reference to tools that were not available that were generally available in at least one world but not in the ones where they wanted to use them: "*Like you might need it [i.e., tools that are unavailable] but you don't have it and it's sometimes confusing if it's there but you can't use it*". Other times, students described the number values used in the after-game tasks as problematic to their engagement in the task: "I *didn't like when [the numbers] got very high—it's harder and also I can't fit it [the created unit fractions] all on the screen*". Finally, students described not being given enough time to engage in the after-game tasks in some classrooms: "*It was a little difficult because [teacher 3, school 1] didn't give enough time [in the game]. That would have helped me*".

Not Bunny. A final subcategory that emerged from the data was students' perceptions of the Bunny character and their propensity to identify with or see themselves as the Bunny in the game. Some students commented that they did not feel a connection to Bunny and, at times, that they felt like an onlooker to Bunny (as opposed to taking on the role of Bunny), felt frustrated that they had to do Bunny's work for them, and also that Bunny talked too slow and sounded like an adult and not a child. One student explained, "*It's like watching a show. It was hard to understand where he was coming from*" while another student commented, "*Yeah, I don't feel like Bunny. Like I do all the work and Bunny doesn't really show us that much*". Other players agreed and commented that they felt like an onlooker due to the narrative aspect of the game:

> *I kind of connected, I guess because the workers were asking the Bunny to do it. And then it was basically you being the bunny, b/c you had to do the work that the worker asked the Bunny to do, but then again, I don't because for example if I was playing a video game and I was the character it would be like I would be the one talking and there wouldn't be an extra character. I don't know how to explain it, if you get what I'm saying. Because there were like two characters talking. You weren't the one who was actually talking.*

4.2.3. Dreaming Bigger

Alongside the areas where students had difficulties relating to the program, students also had suggestions for improvement. Subcategories included (a) Agency, (b) Dynamic interactions, and (c) More of "me". Each is explained below.

Agency. Students offered several suggestions for how to include additional options into the program. For example, students made suggestions about additional content (math/subject topics, extra games, quizzes, etc.) that could be included in the game. For example, some students requested that the game have 'sub-games' available within it where they could go to engage in related skills and concepts. Related to this, students made suggestions to give players the opportunities to talk with the STEM coaches and hold real-time conversations with them as opposed to having feedback statements pop up. For example, one student noted, "*Like type in what you want to say. We could talk to them! Like have something we can press that have little sentences that we could talk to them. And then they will talk to the other person doing your job and just keep talking and get to the questions*". Another student suggested an interactive notepad where students could note strategies and aspects of problems that seemed invariant across challenges: "*I was going to say maybe you could click the notepad and if you could come up and you could try to figure it out on your own. And if you figure it out then you could go back & do it how you just did it. And if needed you could go back and try again*". These suggestions spoke to students wanting more agency in their gameplay.

Dynamic interactions. Students also made suggestions for how certain core components of the program were presented. The first area was the types of prompts or hints to provide players in the game and/or how/when they appear. Three students' ideas appear below. One of the comments relate to students wanting different formats for the prompts, one relates to wanting different levels of prompts, and the last relates to wanting conversation with the coach characters in each level as a form of a prompt:

> *[Student 1] Yeah, like if you weren't paying attention, I'm not saying you cannot like, like the thing was too small for anyone to really see it and they weren't focusing on the thing in the corner . . . so hints won't just pop up and you won't get immediate help. Maybe you could have to press the character, the person—like for the wind turbine level, the wind technician, the bunny, they're all bunnies—you could press and it could give you a really useful hint.*

> *[Student 2] Maybe instead of us having to read it there could be a pop-up screen where the prompt would auto read to you. Or there could be different levels of prompts . . . like you could click something and it could be a video hinting how to partition it and stuff. Like not giving you the answer but helping you partition.*

> *[Student 3] Maybe we could talk to them! Like type in what you want to say . . . Bunny would be talking to them and we would have to listen to what they're saying, maybe we could be Bunny and we could talk to [the STEM coach], and they could answer us.*

Another area for suggested change was the nature of the tools and how they could be used. One student suggested a tool that could minimize dragging, which seems to be a repetitive, time-consuming act in one of the worlds: "*Or like maybe if you did one of them, if you cut the pieces up and every single one and you partitioned it into one the computer will do the rest of it for you*". Other students wanted to see change in how answers in game challenges or after game tasks were presented:

> *Yeah, I was thinking maybe—some people when they click on the wrong answer, they just start clicking on different ones, not really thinking about it. Maybe you can—you could when it's wrong click on the other ones—maybe you could type it in so they actually pick it and if it tells you, it's wrong, they have to write it out—could help them understand more.*

Other students suggested that problems could be adaptive to their gameplay and be simpler or more difficult based on players' strategies. Additionally, other students wanted more agency in how to navigate the game by being able to work on different levels

at different times as opposed to a linear progression. Finally, students suggested that after-game tasks that involved worked examples could instead ask students to recreate strategies they used in actual gameplay. All suggestions speak to students wanting changes in terms of how core components of the program were designed or presented.

More of "me". A final suggestion in this subcategory centered on humanizing aspects of making the narrative of the game and the prompts/hints more relatable. One student explained, "*I would say to give him actual human feelings. Like if he was talking to Eli and he said, 'can you move this' and I'm like, 'sure but how do I do it?' It will make it more interesting*". Another option that students suggested was to encounter or create different characters in the game and to hear their own voices as the Bunny character. One student commented, "*I would like to be able to put my own voice in b/c I know my voice better*". Another student suggested making the characters more emotive: "*Like [if they could show] frustration, or sadness, or even just monologuing to yourself*". These suggestions potentially speak to students wanting to see more emotion and/or more of themselves in the game.

### 4.3. Data Merging and Interpretation

Percentages of code in each category and subcategory by school along with engagement cluster, fraction, and STEM effects are listed in Table 4.

**Table 4.** Percentages of Category/Subcategories by School, Fraction/STEM Changes, and Engagement Cluster.

| | Pre-Post Tests & Engagement | | | Focus-Group Categories/Subcategories | | | | | | | | |
| School | Fraction Effects | STEM Effects | | Engage Cluster | Accessible, Enjoyable Learning | | | Can't Relate | | | Dreaming Bigger | | |
| | | Math Attitudes | Interest in STEM | | Access | Enjoy | Promote | Not Math | Not What I Need | Not Bunny | Agency | Dynamic | More ME |
| 1 | 0.32 Small | -- | -- | Low | | 21.4% | | | 45.4% | | | 33.3% | |
| | | | | | 1.1% | 6.9% | 13.4% | 31% | 4.8% | 9.5% | 13.7% | 8% | 11.6% |
| 2 | 0.49 Medium | -- | -- | High | | 39.6% | | | 38.2% | | | 22.2% | |
| | | | | | 5.6% | 9.7% | 24.31% | 8.6% | 5.2% | 24.3% | 8% | 5.9% | 8.3% |

Of the students of teachers in school 1, 21.4% of the coded qualitative talk turns indicated Accessible, Enjoyable Learning, 45.4% indicated Can't Relate, and 33.3% indicated Dreaming Bigger. Conversely, students of teachers in school 2 had 39.6% of their coded qualitative talk turns indicate Accessible, Enjoyable Learning, 38.2% indicated Can't Relate, and 22.2% indicated Dreaming Bigger. Examining trends across the subcategories reveals different trends within each school that connect with the quantitative results repeated earlier.

Specifically, within School 1, the subcategories Access, Enjoy, and Promotes My Learning (Accessible, Enjoyable Learning) appear in 1.1%, 7%, and 13.4%, respectively, of the coded talk turns across students with school 1 teachers. The subcategories Not the Math I am Used To, Not What I Need When I Need It, and Not Bunny (Can't Relate) appear in 31%, 4.9%, and 9.5%, respectively, of the coded talk turns across students of teachers in school 1. Together, these data add a "why", or a context, for the low engagement, small effect sizes for fraction scheme changes, and the negative, small effects on these students' Interest in STEM. For example, the qualitative trends of math in the game-enhanced program not being what students know as mathematics may specifically connect with their low engagement and modest scheme changes. Additionally, the modest yet negative changes in STEM interest may be linked to the high percentage of these students' requests for more agency or not identifying as the main character in the game.

Conversely, within School 2, the subcategories Access, Enjoy, and Promotes My Learning (Accessible, Enjoyable Learning) appear in 5.6%, 9.8%, and 24.31%, respectively, of the coded talk turns across students with school 2 teachers. The subcategories Not the Math I am Used To, Not What I Need When I Need It, and Not Bunny (Can't Relate) appear in only

6.3%, 10.4%, and 12.5%, respectively, of the coded talk turns across students of teachers in school 2. Together, these data add context, or reasons, for the high engagement, medium effect sizes for fraction scheme changes, and the null effect on students' STEM Interest. For example, the qualitative trends of the program as 'promoting my learning' may specifically connect with these students' encouraging changes in fraction schemes. However, the lack of positive changes in STEM interest may be linked to the high percentage of these students not identifying as the main character in the game.

Looking across the focus-group data for the two schools provides not only context for the quantitative results but reveals important differences, or different pictures, of how the game-enhanced fraction curriculum impacted students' engagement, fraction schemes, and STEM interest in each school. For example, the average number of talk-turn codes within the Enjoyable, Relatable Learning category was significantly different between students of teachers in school 1 and students of teachers in school 2; $t = -5.158$, $p < 0.001$. Moreover, while the average number of talk-turn codes within the Can't Relate category was not significantly different between students of teachers in school 1 and students of teachers *in school 2* ($t = 0.379$, $p > 0.05$), students of teachers in school 1 had far more "Not the math I am used to" subcategory codes. These data are consistent with the lower engagement and smaller fraction scheme changes found in the quantitative data for school 1. Therefore, we conclude that the program positively impacted students in both schools in terms of the fraction scheme changes, but students in school 2 perceived their learning in the game-enhanced supplemental fraction curriculum as more accessible, enjoyable, and having promoted learning more than students in school 1.

## 5. Discussion

The purpose of this study was to determine how a game-enhanced, universally designed fraction intervention impacts outcomes for students with and without HID. Specifically, we addressed the problem of how a game-enhanced supplemental program impacts students' fraction knowledge, engagement, and STEM interest. Prior research suggests the potential of game-enhanced learning environments to enhance STEM content accessibility, increase problem solving, and improve students' ability to build, internalize, and communicate knowledge [32–35]. Results of this study add to this literature in documenting positive impacts of the program specifically on students' access to and internalization of fraction knowledge. That is, students' fraction schemes as demonstrated by their significant changes in score were consistent across the two schools. The small to medium effects on students' fraction learning reported in this study suggest the potential of the program to promote outcomes in a foundational STEM concept for historically marginalized students included in our study, such as students with HID and intersecting identities [69]. Thus, the supplemental game-enhanced fraction program warrants future research with respect to whether positive effects on students' fraction learning are impactful for these students beyond the bounds of this study.

At the same time, the results of this study show different patterns of students' self-reported engagement in the program. Specifically, the data analysis revealed two different engagement clusters across the program as a whole; one with students self-reporting high engagement and another with students self-reporting low engagement. Additionally, students in school 2 were significantly more likely to be members of the high-engagement cluster, while students in school 1 were significantly more likely to be in the low-engagement cluster. Moreover, both measures of STEM interest (i.e., math attitudes and interest in STEM careers) showed no significant change for students in school 2 and only small, negative changes for interest in STEM, specifically, for students in school 1. Qualitative data produced contextual backing that suggests the "why" behind these quantitative findings when these were uncovered and later quantified and merged with the other data sources.

On the one hand, students in school 2 reported their access, enjoyment, and learning in the program in the focus groups at a level that was significantly greater than students in school 1. On the other hand, while not significantly different, students in school 1 reported

that they had difficulty relating to the program because they did not view mathematics learning as occurring through their own problem solving (without first being shown by a teacher) or through their own actions more often than students in school 2. Furthermore, students in school 1 reported wanting more agency in gameplay, and both schools reported not being able to take on the role of the main character in the game. Prior research suggests that the agency students with HID have over the ways in which they learn is necessary to promote cognitive change [7,30,44,69]. Findings from this study support and extend prior research in that students who reported feeling like they had less agency in the game and who could not relate to the game character and the mode of content learning reported a decreased change in their interest in STEM and low engagement in all game worlds. These findings suggest a possible link between student perceptions of agency, their sense of identifying with characters, self-reported levels of engagement, and STEM interest.

The need for increased agency in the game may also point to self-determination. Self-determination theory states that all humans have three basic psychological needs—autonomy, competence, and relatedness—that underlie growth and development. Autonomy refers to feeling that one has choice and is willingly endorsing one's behavior. It may explain why promoting autonomy/agency was such a pressing need for the students in terms of their perceptions of having to take on a particular character whom they struggled to relate to/identify with. Gaming can promote self-determination, but the ability to do so is contingent on the characteristics of the game [70]. Thus, self-determination may be something that was underdeveloped in the video game or the curriculum more generally.

Taken together, the findings suggest that while the core components of this game-enhanced supplemental fraction program can positively impact learning, revisions are necessary to increase engagement and STEM interest for all students. For example, the role of agency, self-determination, and students' propensity to identify as the character deserves more attention in our work. Additionally, the cognitive prompts, which are a core component of our program [45,46,50] were heavily mentioned in the Dreaming Bigger category in terms of how students experience prompts in the game. Digital games including immediate cognitive feedback improved students' mastery of problem-solving tasks within a game. Students across the two schools in our study requested to have videos, real-time statements about their gameplay, or to be able to interact in real-time conversation with the STEM coaches in each world. Interactive delivery of game prompts may better align with supporting students' noticing and reflecting upon their actions in the game. Promoting noticing and reflection in students' thinking while solving problems is underutilized in instructing students with HID [51–53]. Interactive components may also provide opportunities in the game for students to feel better connected to the characters and the context through the use of emotion (e.g., emotive expressions, voicing excitement or frustration). Finally, it is interesting that school 2 had larger learning effects and more engagement, as they used a different after-game task as a part of the game-enhanced supplemental program. Specifically, students and their teachers in School 2 used game replays, while students and teachers in School 1 used worked examples. Arguably, game replays could create more opportunities for students to use their own reasoning and have a sense of agency because they ask students to re-enact and discuss strategies they used in gameplay as opposed to critiquing a pre-conceived solution. It is an empirical question whether the program with game replays versus the program with worked examples results in greater agency, engagement, or learning for students. Future research should test this idea.

## 6. Limitations and Future Research

This study has limitations that need to be addressed in future research. First, due to the small sample size and use of qualitative data, the findings are limited to this group of students only. Second, the engagement data is self-reported and was collected after each game world, so the data is removed from students' real-time gameplay and conversations within each program lesson, which may suggest that it is not specific to particular learning

events. A third limitation rests in the lack of a comparison group. This study was part of a larger study on the program's feasibility in elementary school settings and was not a test of the program's effects compared to business as usual. As a result, we do not yet know if the results reported within are attributable to the supplemental intervention. Finally, the role of teacher implementation and the norm match/mismatch of the regular classroom are not clear in this study. In focus groups, students gave feedback on the program and its components, but it is not clear the degree to which teachers implemented the program with integrity or the extent to which aspects of program adherence or teacher quality impacted student outcomes [71]. Although elementary teachers are enthusiastic users of mathematical games, they tend to have a strong preference for using non-digital games in their classroom compared with digital games [72]. Teachers might not have been enthusiastic about focusing mathematics learning by using a computer game.

Future work should examine the role of teacher adherence and their views of the curriculum with respect to student perspectives of the program—specifically, students' view that the mathematics they encountered in the game-enhanced supplemental curriculum is not the math they are used to in their regular math classrooms or are an alteration of the program's core components. Future work could also examine the relationship between observed teacher enthusiasm/quality and students' reported engagement and STEM interest.

## 7. Conclusions

In this work, we learned that a game-enhanced supplemental curriculum has the potential to positively impact students' fraction knowledge and schemes. Program revisions are planned to produce increased impact engagement and STEM interest, specifically in relation to how students see themselves as the character, the opportunities for self-determination, the number of choices they have in the game, and the ways in which they receive support in the game. Given the continued need to produce STEM-prepared and -interested students, the need for continued work and research is paramount.

**Author Contributions:** Conceptualization, J.H.H.; methodology, J.H.H., A.D., B.B. and M.T.; software, J.H.H. and M.M.; validation, B.B. and A.D.; formal analysis, J.H.H., A.D., B.B. and M.T.; investigation, J.H.H., A.D. and B.B.; resources, J.H.H., M.M. and M.T.; writing—original draft preparation, J.H.H., M.M. and M.T.; writing—review and editing, J.H.H., K.H. and M.T.; visualization, A.K.; supervision, J.H.H., M.M. and M.T.; project administration, J.H.H.; funding acquisition, J.H.H., M.M. and M.T. All authors have read and agreed to the published version of the manuscript.

**Funding:** This research was funded by US National Science Foundation, grant number 1949122. Views expressed in this paper are those of the authors and do not necessarily reflect those of the National Science Foundation.

**Institutional Review Board Statement:** The animal study protocol was approved by the Institutional Review Board (or Ethics Committee) of The University of Central Florida (protocol code 14238A03, 11 December 2021).

**Informed Consent Statement:** Informed consent was obtained from all subjects involved in the study.

**Data Availability Statement:** Data are available by request from the authors.

**Conflicts of Interest:** The authors declare no conflict of interest.

## Appendix A

Dream2B Student Focus-Group Protocol and Questions
\# of participants: \_\_\_\_\_\_\_\_\_\_\_\_\_\_\_\_\_\_\_\_\_\_\_\_\_\_\_\_\_\_\_\_\_\_\_\_\_\_\_\_\_\_\_\_\_\_\_\_\_
Host: \_\_\_\_\_\_\_\_\_\_\_\_\_\_\_\_\_\_\_\_\_\_\_\_\_\_\_\_\_\_\_\_\_\_\_\_\_\_\_\_\_\_\_\_\_\_\_\_\_\_\_\_\_\_\_\_\_\_\_
Date: \_\_\_\_\_\_\_\_\_\_\_\_\_\_\_\_\_\_\_\_\_\_\_\_\_\_\_\_ Site: \_\_\_\_\_\_\_\_\_\_\_\_\_\_\_\_\_\_\_\_\_\_\_\_\_\_\_\_\_\_
Welcome and thank you for agreeing to participate. My name is \_\_\_\_\_ and I am here with \_\_ and \_\_\_\_. We will be leading today's focus-group discussion. We're interested in finding out about your experience in the ModelMe program, which included the Dream2B

game and fraction lessons. I'll be asking you some questions to learn more. Do you have any questions about your participation in this group today? Notes will be taken during the focus-group discussion; however, we will not record your name. The discussion will take 20–25 min. This discussion will be recorded and transcribed (that means what people say will be written in a document) later by us. Do we have your permission to record the conversation? Wait for everyone to say yes and then switch on the recorder.

The purpose of this focus-group discussion is to get your honest opinions about the ModelMe program and Dream2B game. This is not a test and there is no grade for our time together today. There are no wrong answers. I'll ask you the questions; ___and ___ are writing down notes as we talk. If you don't want to continue to talk in the focus group today, please let us know.

Are there any questions before we begin?

Goals:

(a)　What are students' perceptions of the Dream2B game and curriculum?
(b)　Are the students having positive experiences with the program? Do they see it as engaging?
(c)　Do students feel that they had a voice in their learning?

Audience: Students
Questions for Student Focus Group
What were your overall impressions of the game?

- What did you like and/or not like about the game?
- What would you add and/or remove from the game?
- What did the game help you to learn (or teach you) about fractions?
- Did you feel like you became Bunny when you played the game? What helped you feel that way?

○　Listen for things like being able to choose the hair, eyes, etc. for the Bunny or identifying with Bunny

How did the game help you learn and solve fraction problems?

- What was the hardest thing to figure out how to do in the game?
- Was there anything that took a while for you to know how to do?
- How did the tools help you in the game?
- How did the hints help you in the game?
- Were different worlds easier or more/less fun than others?

What did you like or not like about the lessons after the game?

- How did they help you complete the tasks in the game?
- What was easy/hard about completing the after-game tasks?

What did you like or not like about the launch videos?

- Did the videos help you understand the careers better?
- Did the videos make you want to know more about the career? Which one(s)?

Is there anything else you want to share about the game or the curriculum?
Probes:
Can you tell us more about that?
Can you be more specific?
Do you have any examples?
Do any of you have similar experiences?
Again, thank you for your participation.

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
