# Peer review of "Effects of Game-Enhanced Supplemental Fraction Curriculum on Student Engagement, Fraction Knowledge, and STEM Interest"

_education, doi:10.3390/educsci13070646_

Round 1

Reviewer 1 Report

The study deployed a sequential mixed methods design to investigate how a game-enhanced intervention impacted students' fraction knowledge, engagement, and STEM interests. The method is sound and the result is relevant in the STEM learning. 

Author Response

Thank you for your positive reviews!

Reviewer 2 Report

The article deals with the underrepresentation of individuals with disabilities in STEM as well as ICT. It points to insufficient materials suitable for pupils with disabilities in primary school, with which they could demonstrate their knowledge and skills.

The authors used a sequential mixed methods design to show how a game-enhanced intervention affects students' partial knowledge.

The research was conducted in two American elementary schools, with a sample size of 133 students, and the sample included a variety of races, languages, and disabilities. The research took place over 9 weeks. The teachers completed four half-day training sessions on the implementation of the supplementary curriculum.

Quantitative results revealed statistically significant effects of the program on students' understanding and engagement, but not on their interest in STEM.

The work is with a good logical structure. Methods of processing the work, i.e. mainly the study of literature, analysis, synthesis, comparison and experiment within the empirical part are chosen correctly with regard to the objectives, subject and scope of work. The work has good form, style and language level. There are 3 tables, 4 graphs and 53 literature sources (journals, books and also online sources) used in the work, which are correctly cited in the text. There are just small text errors (e.g. in table header, p. 18 etc.). Conclusion part could have been longer. The shortcoming of the work, as the authors themselves state, is the small research sample as well as the impossibility of pointing out the direct connection of the study, as it was part of a larger study. Therefore, further research is recommended, mainly on a larger sample of monitored pupils, as well as preventing the influence of other research. It is not entirely clear from the study to what extent the teachers implemented the program and to what extent it matched/did not match the norms in the regular classroom. It would be appropriate in further research to examine the specific impact of mathematics supplemented with games on student interest, as well as the impact of teacher enthusiasm and student engagement and interest in STEM.

In general, this work very well suits to the journal's scope and can be used as a suitable background for further research. I recommend to accept this study after minor revisions (mentioned above).

Thank you. Kind regards.

Author Response

Thank you for your positive reviews of our work.  Below are a few responses to your helpful comments (our response is italicized):

There are just small text errors (e.g. in table header, p. 18 etc.). Conclusion part could have been longer.

Thank you for making us aware of small text errors.  We have corrected them in the updated version.

 The shortcoming of the work, as the authors themselves state, is the small research sample as well as the impossibility of pointing out the direct connection of the study, as it was part of a larger study. Therefore, further research is recommended, mainly on a larger sample of monitored pupils, as well as preventing the influence of other research. 

Thank you for agreeing the stated limitations of the work.  We agree and the information is in the limitations section as noted.

It is not entirely clear from the study to what extent the teachers implemented the program and to what extent it matched/did not match the norms in the regular classroom. 

Thank you for agreeing with the stated implications and future directions of this work.  We agree and the information is in the implications section as noted.

It would be appropriate in further research to examine the specific impact of mathematics supplemented with games on student interest, as well as the impact of teacher enthusiasm and student engagement and interest in STEM.

Thank you for agreeing with the stated implications and future directions of this work.  We agree and the information is in the implications section as noted.

Reviewer 3 Report

Thanks for the opportunity to review this very clearly presented and comprehensive study on how a game-based program could support student understanding of fractions. I was impressed by all components of the manuscript, including the literature review, the detailed description of the methods, intervention and analysis, the presentation of the data, and the discussion/ conclusions. 

Two points that might be worth considering. First, I wonder whether referencing self-determination theory in the discussion in order to explain why the intervention was perhaps less effective than it might have been, is worth considering. Self-determination theory can for example explain why promoting autonomy/ agency was such a pressing need for the students and perhaps something that was under-developed in the video games, where they were forced to take on a particular character who they struggled to relate to/ identify with. The notion that gaming can promote self-determination, but that this is contingent on the characteristics of a game, has been argued here: Rigby, S., & Ryan, R. M. (2011). Glued to games: How video games draw us in and hold us spellbound: How video games draw us in and hold us spellbound. AbC-CLIo.

The second point is perhaps more minor, but also might explain the limited impact of the study in School 1 (potentially). There is evidence that, although elementary teachers are enthusiastic users of mathematical games, they tend to have a strong preference for using non-digital games in their classroom compared with digital games (see https://www.iejee.com/index.php/IEJEE/article/view/1302). This is perhaps something to ponder when you note in your limitations section that the extent to which teachers implemented the program as intended is unknown, and may have impacted on the findings (e.g., you might speculate that teachers might not have been enthusiastic about focussing mathematics learning around a computer game).

Thanks again for a great article. I will no doubt refer to it once it is published. 

Author Response

Thank you for your positive review of our paper.  We really appreciate your helpful suggestions and respond to each below (our responses in highlighted italics):

Two points that might be worth considering. First, I wonder whether referencing self-determination theory in the discussion in order to explain why the intervention was perhaps less effective than it might have been, is worth considering. Self-determination theory can for example explain why promoting autonomy/ agency was such a pressing need for the students and perhaps something that was under-developed in the video games, where they were forced to take on a particular character who they struggled to relate to/ identify with. The notion that gaming can promote self-determination, but that this is contingent on the characteristics of a game, has been argued here: Rigby, S., & Ryan, R. M. (2011). Glued to games: How video games draw us in and hold us spellbound: How video games draw us in and hold us spellbound. AbC-CLIo.

Thank you so much for this recommendation.  Self-determination theory is an interesting explanation for the results and we have now incorporated it into our discussion.

The second point is perhaps more minor, but also might explain the limited impact of the study in School 1 (potentially). There is evidence that, although elementary teachers are enthusiastic users of mathematical games, they tend to have a strong preference for using non-digital games in their classroom compared with digital games (see https://www.iejee.com/index.php/IEJEE/article/view/1302). This is perhaps something to ponder when you note in your limitations section that the extent to which teachers implemented the program as intended is unknown, and may have impacted on the findings (e.g., you might speculate that teachers might not have been enthusiastic about focussing mathematics learning around a computer game).

Thank you again for this second helpful suggestion.  We agree and now include this point and reference in our discussion of the results, especially for school 1.  It is particularly interesting because school 2 did tend to use more technology regularly than school 1 (although this is anecdotal).  We appreciate this suggestion!